# A Novel Method of Supporting the Laser Welding Process with Mechanical Acoustic Vibrations

**DOI:** 10.3390/ma13184179

**Published:** 2020-09-20

**Authors:** Arkadiusz Krajewski, Grzegorz Klekot, Marcin Cybulak, Paweł Kołodziejczak

**Affiliations:** 1Faculty of Production Engineering, Warsaw University of Technology, 02-524 Warsaw, Poland; cybulak.marcin@gmail.com (M.C.); pkolodzi@wip.pw.edu.pl (P.K.); 2Faculty of Automotive and Construction Machinery Engineering, Warsaw University of Technology, 02-524 Warsaw, Poland; grzegorz.klekot@pw.edu.pl

**Keywords:** laser welding, mechanical acoustic vibrations, structure properties

## Abstract

The research described in this article presents a new contactless method of introducing mechanical vibrations into the base material during CO_2_ laser welding of low-carbon steel. The experimental procedure boiled down to subjecting a P235GH steel pipe with a 60 mm diameter, 3.2 mm wall thickness and 500 mm length to acoustic signals with a resonant frequency during the welding process. Acoustic vibrations with a frequency of 1385, 110 and 50 Hz were introduced into the pipe along the axis and transversely from the outer surface. The obtained welds were then subjected to structural tests and Vickers hardness measurements. The results of comparative tests show the impact of such introduced vibrations on the granular structure of the welds, as well as on their microhardness in specific areas, such as the face, penetration depth and the heat-affected zone. The effectiveness of the proposed method of introducing vibrations in the scope of grain size and shape as well as changes in the hardness distribution in the obtained welds is demonstrated.

## 1. Introduction

In recent years, increased interest among researchers in finding new ways of improving the structure [1,2] and mechanical properties of welded joints or coatings [3] applied by welding methods [4,5,6,7], as well as improving the quality of metal alloys [8,9], can be observed. The issue of improving the micro and macro structural properties can generally be interpreted as improving metallurgical weldability, which has often been a priority [10]. A careful observer of the literature on the subject will notice that the majority of research in which the authors used mechanical vibrations focuses on the effects obtained, which undoubtedly leads in many cases to the fragmentation of grains [11], changing their shape into more equiaxial forms [12,13], improving mechanical properties such as strength, impact strength or hardness [14,15,16]. Few of these studies attached significant importance to technical problems associated with the effective introduction of vibrations and their form [2,17], and even more so with other parameters such as frequency, amplitude or phase [1,2,12,18].

Some researchers, such as [1], investigated ultrasonic-assisted laser welding on stainless steel, which was carried out to find out the effect of ultrasonic energy input on the microstructure and the strength of the weld metal. It is well known that the energy of ultrasonic vibrations depends mainly on the amplitude. It is also known that the effective transfer of vibration energy is not easy and depends on the rigidity of the attachment of the vibration system to the welded object. In this case, however, it would be difficult to expect a homogeneous distribution of energy, as shown by the laser vibrometer used for measurements [1]. As could be expected, the energy value (amplitude) strongly depended on the vibration phase in the object [2,18] and the obtained results were conditioned by the occurrence of an arrow or vibration node. Therefore, the use of such methods of introducing vibrations whose energy varies with the distance of the source of vibrations from the liquid metal pool does not guarantee repeatable conditions of the technological process [1,2,5,10,12,18].

On the other hand, full control of mechanical vibrations is only possible when they are introduced by a well-tested and rigidly mounted vibrating system [2,9,13,14,15,18,19,20] or modulation of the electric arc [21,22,23] or laser beam [3]. The improvement of structural features and mechanical properties is obtained in various zones of welds or coatings, i.e., in the weld itself and in the heat-affected zone. Nothdurft in [24] proves during welding dissimilar materials the width of the unmixed zone between them was reduced due to the micro-turbulence formed near the fusion boundary caused by the cavitation effect. With the use of ultrasonic laser excitation, the main crystallography textures in the weld metal are shifted, and the ratio of misorientation was significantly improved. Quality improvement is achieved in many ways. These may be actions aiming at the optimization of technological parameters of the used welding methods or additional procedures using heat treatment, hybrid methods and innovative ways of affecting the liquid metal pool with various physical factors, such as an electromagnetic field or mechanical vibrations. Of these, it is worth paying attention to the latter, despite the fact that they do not occur very often and do not unconditionally guarantee success in every case of welding, surfacing or remelting. The use of mechanical vibrations carries a number of problems and limitations, depending on how they are introduced into the basic material, and raises some manufacturers’ fears, caused mostly by ignorance or lack of access to complicated vibration generating devices, but also technical difficulties related to ensuring steady contact between the basic material and the vibrating system [2,18]. All this means that the manufacturer of welded structures may not decide to use mechanical vibrations as the factor that, when properly used, will allow them to achieve the assumed quality improvement.

The application of mechanical vibrations during welding processes shows that satisfactory results can be achieved without the need for direct and reliable attachment of the vibrating system to the base material [2,12,18,25]. Vibrations can be introduced, for example, by means of laser beam pulses [3,17], controlled modulation of the electric arc or impact methods using mass collision energy transfer [6,25]. An interesting technical solution was used by researchers [26] during TIG welding of aluminum alloys. They used an ultrasonic waveguide in the form of a roller and by rotating it were able to maintain a constant distance of the contact point from the molten metal pool. This resulted in an increase in hardness of over 8%, and an increase in tensile strength by almost 30%. The contact method of delivering ultrasonic vibrations in underwater welding in [27] also caused changes in mechanical properties. (increase in average hardness, toughness and tensile strength, grain refinement and reduction of the width of the boundaries between ferrite grains). This effect was achieved by the solid contact of the waveguide with the welded material.

One of contactless methods is presented in this article. It is interesting because vibrations are introduced by the acoustic route, meaning that, first of all, this process is contactless, and secondly, it provides many possibilities of modification of the parameters characterizing mechanical vibrations, such as frequency, amplitude or distance from the welding heat source. Among the research works, there are various technical solutions with specific methods of introducing vibrations into objects subjected to welding processes. These include vibrations generated by tables on which the base material is mounted [7,16], or, during arc welding, vibrations that are carried by a consumable electrode [5,13,22]. All methods of supplying vibrations to the base material are worth attention, although not all of them can be used in any conditions of access to the material or when the distance from the heat source is too close to allow it. There are some original technical solutions regarding the introduction of vibrations by various methods, which are the subject of patent studies [6,17], but there are no such ones that provide the possibility of introducing vibrations without providing direct contact and fixing the vibrating system with the base material, which is a major impediment in industrial conditions. The impact of vibrations through carriers not related to the basic material gives a chance to avoid problems related to the proximity of the heat source and to simplify the technical process of introducing vibrations. Therefore, it is important to propose such a method of introducing vibrations that does not require direct contact with the base material. One such method was used in the research described in this article. It allows for easy application in industry and for the economical conditioning of structures obtained in welding processes.

## 2. Materials and Methods

The study used a P235GH steel pipe according to PN-EN 10217-2 with a diameter of 60 mm, wall thickness 3.2 mm and length 500 mm (the composition and properties of this steel are listed in Table 1).

Before the welding tests were performed, the resonant frequency of the pipe was determined along its axis (*x*) and in the transverse direction (*y*). During the measurement, the pipe was hung, and the measuring sensors were attached halfway along its length and at one of its ends. A hammer drill, three piezoelectric accelerometers, and PULSE analyzer environment with the Bruel and Kjaer type 3050-B-060 module (Bruel and Kjaer, Naerum, Denmark) were used for the tests (Figure 1).

In order to test the natural frequency, the pipe was hung loosely on a thin rope, i.e., in conditions of the minimum number of ties, in isolation from friction (Figure 1), and was stimulated by impulse stroke. Then, the pipe vibrations were recorded in two mutually perpendicular directions in the frequency range from 0 to 6 kHz. The methodology of vibration generation using the acoustic method and the devices, sensors and software used are described in more detail in [28,29]. In order to eliminate the instability of most arc welding methods during vibration process, it was decided to use a laser source, which, by its nature, guarantees the stable and repeatable welding of the substrates in these conditions. Welding tests were carried out in a flat position parallel to the pipe axis. Weld penetrations were obtained: Without the participation of mechanical vibrations, and with the participation of acoustically generated mechanical vibrations. A CO_2_ laser with fast longitudinal flow of VFA2500 type from Wegmann–Baasel (Aschheim, Germany) was used for welding. It can work in pulse or constant mode in the power range from 100 to 2500 W. The radiation wave generated by the laser is 10.6 µm long. A mixture of CO_2_, N_2_ and He gases was used as protection. In the experiments, a helium blast in the beam axis was used with a capacity of 10 L/min through a nozzle 4 mm in diameter spaced from the parent material surface by 5 mm. A beam of coherent light was used, which was 1 mm in diameter on the surface of the workpiece and in the TEM 10 mode. As the focus (d = 0.4 mm) of the beam fell above the surface, it was impossible to obtain a steam channel. This was dictated by the desire to obtain a larger size of the metal pool, thus allowing for easier observation of changes in its structure after crystallization. As a result, linear welds with a length of about 150 mm arranged parallel to the pipe axis on its outer surface were obtained: Weld No. 1, with vibrations at a frequency of 50 Hz introduced transversely to the axis and perpendicular to the surface of the pipe; weld No. 2, with vibrations at a frequency of 50 Hz introduced along the pipe axis to its interior; weld No. 3, with vibrations at a frequency of 1385 Hz introduced along the pipe axis to its interior; weld No. 4, reached without vibrations; and weld No. 5, with vibrations at a frequency of 110 Hz introduced along the pipe axis to its interior. Figures 3 and 4 show the two ways of introducing acoustic vibrations—transversely and longitudinally—in relation to the pipe axis, respectively. Laser welding parameters and specifications regarding the introduced vibrations are presented in Table 2.

For the generation of acoustic vibrations, the Bruel and Kjaer type 1023 harmonics generator with the 2734 amplifier (Bruel and Kjaer, Naerum, Denmark) was used, as well as the Bruel and Kjaer type 4295 sound source. The Bruel and Kjaer type 2236 m was used to measure the sound pressure level. In welding tests, sound signals with a constant sound pressure level of approximately 100 dB (effective value of the acoustic pressure amplitude is about 2 Pa) were used.

After performing experimental welding tests, five longitudinal welds were obtained, which were subjected to comparative metallographic tests and Vickers hardness measurements at a load of 100 g.

Metallographic tests of the samples were carried out with an OLYMPUS BX51M optical microscope (Olympus Optical Co., Ltd., Tokyo, Japan) with Stream Essential v. 2.3 software (Olympus Optical Co. Ltd, Tokyo, Japan) and an electron microscope JEOL JSM-7600F (JEOL Ltd., Tokyo, Japan) to obtain images on the cross-section of welds and the surface distribution of elements.

## 3. Results

Tests of the geometry, structure and properties of the obtained welds were preceded by determining the resonance frequency of the welded object, which was the pipe. Then, optical and electron microscopy was performed, and granular structure analysis, weld geometry measurements and elemental concentration distribution were examined. The results of these studies are presented below.

### 3.1. Resonance Frequency Determining

The first important research step was to determine the resonance frequency of the pipe undergoing the welding process. Stimulation with vibrations in the resonance area results in a high amplitude of vibrations, a particular gain occurs when the forced frequency is equal to the characteristic resonance frequency of the object. In the case of welding, the effect of using vibrations with a resonant frequency may not give the best result in terms of structure and mechanical properties. However, determining the resonance frequency will be treated as a reference point in these studies. The obtained results of the experiment carried out to determine the natural vibrations are presented by the vibrations spectra in Figure 3a,b.

Analyzing the course of the graphs of the natural vibrations amplitude for the vibrations frequency value, it can be clearly seen that the resonant frequency determined on the basis of vibrations in the *x-* and *y*-axis is about 1385 Hz.

### 3.2. Shape Factors of Welds

Geometry parameters were performed on cross-sections of the obtained weld penetrations in half of their length. A magnification of 200× was used to show significant differences in the microstructure of individual joint penetration areas. Metallographic images of welds allowed us to assess the width of the face *B*, depth of penetration *H* and allowed to calculate the shape factor *B*/*H*. All listed geometrical features were determined based on three samples obtained with a specific set of technological parameters and are presented in Table 2, while shape parameters are presented in Table 3 and Figure 4. The results regarding the shape factors are the arithmetic mean of the three cross sections. In the course of metallographic tests, the differences in the measured values were within the range of +/− 10 μm

It is noteworthy that in general, with the use of vibrations, the width of the face decreases in relation to welding attempts without the use of vibrations. This effect is most clearly visible in the case of vibrations with a frequency of 50 Hz introduced along the axis of the pipe (weld No. 2). The depth of penetration in this case increases the most. One can also observe a correlation with the *B*/*H* shape factor, which reaches the minimum in the test No. 2. An interesting observation is that the introduction of vibrations with a frequency of 50 Hz transversely to the axis of the pipe, which took place in test No. 1, caused an increase in *B* and a decrease in *H* in relation to weld No. 2. This indicates a greater efficiency of vibrations introduced longitudinally to the welding direction.

### 3.3. Optical Microcopy, Grain Analyses and Hardness Distribution

The structures of the obtained welds were compared at 200× magnification, which allowed us to evaluate the specifics of the grain structure, their directivity and size in accordance with ASTM E 112-13. Figure 5a,c,e,g,i,k contain photos of representative structures of the basic material and those obtained after welding using a laser with the effect of acoustic vibrations of different frequencies and without the use of vibrations. As a result of using the Stream Essential v. 2.3 software, the distribution of grains of a given size was determined and the results are shown in Figure 5b,d,f,h,j,l.

The grain size test was carried out near the face of the obtained joints in the area of a rectangle with sides 105/90 μm, which was marked in the micrographs in Figure 5. The maximum of the grain size curve in the case of weld No. 2 (obtained as a result of the impact of vibrations with a frequency of 50 Hz introduced longitudinally into the inside of the tube) is shifted towards the smaller grains, and therefore the larger grain size number. The course of the remaining curves obtained for welds No. 1 and 5 in Figure 5m is very similar and no major differences can be found.

Hardness was tested on the cross-sections of welds using the Vickers method at a load of 100 g and, as was the case with shape parameters before, one point on the graph was obtained as the arithmetic mean of three measurements. The results obtained are shown in Figure 6.

In the case of weld No. 4 (Figure 6a) obtained without vibration support, a significant increase in hardness can be observed in the area of the heat-affected zone at a depth of 50 μm (approximately 325HV0.1) and a slight increase in the axis of weld metal (250HV0.1). The results of the hardness measurement along the weld axis (Figure 6b) at a depth of about 100 μm showed a hardness of about 300HV0.1.

Weld No. 1 (Figure 6c), which was obtained with a vibration of 50 Hz that was introduced transversely in the heat-affected zones at a depth of 50 μm, did not exceed 240HV0.1, and in the weld metal center, the hardness oscillates around the value of 200HV0.1. Measurements along the weld axis (Figure 6d) at a depth of approximately 200 μm show an increase to 350HV0.1.

Weld No. 2 (Figure 6e), obtained with the use of a 50 Hz vibration introduced longitudinally in the heat-affected zones at a depth of 50 μm, oscillates around 245HV0.1, and in the weld metal center, the hardness does not exceed 270HV0.1. Measurements along the weld axis (Figure 6f) at a depth of approximately 200 μm showed an increase to 350HV0.1.

Weld No. 3 (Figure 6g), obtained with the use of 1385 Hz vibrations introduced longitudinally in the heat-affected zones at a depth of 50 μm, reached about 300HV0.1, while in the weld metal center (Figure 6h), the hardness changed quite significantly from about 160HV0.1 at a depth of 50 μm to, at a distance of 120 μm from the face, approximately 110HV0.1, and then at 200 μm it reached approximately 245HV0.1.

In the case of weld No. 5 (Figure 6i), obtained with the use of vibrations with a frequency of 110 Hz introduced longitudinally in the heat-affected zones at a depth of 50 μm, it oscillated around 250HV0.1, while in the weld metal axis (Figure 6j), the hardness fluctuated quite significantly from the value of about 200HV0.1 at a depth of 50 μm to approximately 110HV0.1 at a distance of 200 μm from the face, and then to approximately 280HV0.1 at 200 μm.

In the case of the weld No. 1, which was obtained with the assistance of vibrations with a frequency of 50 Hz supplied from the outside, perpendicular to the pipe surface, the highest hardness value was recorded at a depth of 200 μm (350HV0.1). This may be the result of the direct impact of cyclic changes in acoustic pressure on the molten metal pool. This process could increase the cooling rate and affect the argon shield at the welding site.

### 3.4. SEM and EDX Examination

In an attempt to elucidate the changes in structure due to the impact of acoustic vibrations, SEM and EDX tests were conducted. The most interesting results are presented in Figure 7.

Interesting changes in the structure were observed by SEM images taken of the face area of the weld center (Figure 7). At a magnification of 2500 times, the biggest difference can be seen in the face zone’s structure between the weld reached within vibration and the corresponding area in the welds obtained with the support of acoustic vibrations with a frequency of 1385 Hz introduced axially. In other cases, the differences are less noticeable, although they do occur. It is clear here that a distinctive area resembling an imbalanced structure was created that does not occur in other cases. The SEM studies in Figure 7a,c,e,g,i show the structures in weld center areas, and those in the face zone are shown in Figure 7b,d,f,h,j. It can be noticed here that when using acoustic vibrations with a resonance frequency of 1385 Hz during welding, the crystallites are clearly arranged in different directions. The same happens when using vibrations with a lower frequency of 50 Hz introduced transversely and axially. When using of acoustic vibrations introduced axially with a frequency of 110 Hz, the change is less pronounced.

As a result of laser welding of the pipe, a thin layer of iron, silicon, aluminum and manganese, about 10 μm thick, forms on the surface. These are probably separations of alloy components of melted steel. In Figure 8, you can see a comparison of the structure of the obtained layers in different samples. It is clear that the face of weld No. 4 (Figure 8a), reached with a vibration-free laser, is characterized by a lack of structure orientation, unlike the other cases. In the case of weld No. 1 (Figure 8b), fused with vibration with a frequency of 50 Hz introduced transversely, and in the case of weld No. 5, obtained with the longitudinal introduction of vibrations with a frequency of 110 Hz, a clear decrease in the silicon concentration in the face can be observed. The case of weld No. 2 obtained with a vibration of 50 Hz (Figure 8c) introduced longitudinally shows that we are dealing with a clear fragmentation of precipitates with a visible orientation here. This orientation of the precipitates perpendicular to the surface of the pipe is most apparent in the case of weld No. 3, obtained with the introduction of vibrations with a resonance frequency of 1385 Hz (Figure 8e).

The results of the SEM and EDX examination of the surface layer of the obtained welds show specific differences in the morphology of the joints fused with a laser and subjected to acoustic vibrations. This is shown in Figure 8.

## 4. Discussion

The presented results show that the acoustic vibrations input can affect both the geometry of the weld and its face’s microstructure. The scale of these changes is not comparable with effects reached in other methods that directly introduce vibrations [1,2,5,10,18,25], but they give the opportunity to reach constant conditions in the welding place using contactless supplying vibration, as in [16,25].

As a result of the conducted tests, the obtained welds did not clearly indicate any tendency. Each one of them deserves attention. Therefore, the support with vibrations of 50 Hz supplied from the outside perpendicular to the pipe contributed to the achievement of the highest hardness in the weld No. 1. The reason could be increased cooling and interaction with the shielding gas.

The maximum depth of penetration and, at the same time, the minimum width of the face were obtained in the case of a weld treated to 50 Hz vibrations spreading axially inside the pipe (weld No. 2). The reasons can be found in the transverse stresses generated by the cyclic change of the sound pressure inside the pipe.

The grain size decreased the most when exposed to vibrations of 50 Hz, propagating along the pipe axis (weld No. 2).

SEM tests, on the other hand, showed the greatest differences in the structure of the top layer for a weld treated with vibrations of resonance frequency. Practically all vibration assisted welds differ in this area from the vibration-free welds. It can be see the directional arrangement of crystallites perpendicular to the surface.

It can be seen that the lower vibration frequency permits the higher weld penetration depth.

The direction of the vibrations introduction is important-when they are introduced transversely to the axis of the weld or axially, a much lower penetration can be observed than in the case of a longitudinal direction.

Supporting the laser welding method with the introduction of acoustic vibrations has a particularly strong impact around the fusion line. Without vibrations, the fusion line is blurred, and it is difficult to determine it clearly; there is also a wide heat-affected zone. With the support of vibrations, the fusion line becomes clearly visible and the heat-affected zone decreases noticeably.

The grain size of the weld supported with vibrations is reduced. The smallest grains were observed in the case of longitudinal introduction of acoustic vibrations with a frequency of 110 Hz.

Hardness measurements showed that the longitudinal introduction of acoustic vibrations generally reduced the hardness value (by approximately 50HV0.1) in the heat-affected zone, regardless of the vibration frequency used. At a resonance frequency of 1385 Hz, the decrease in hardness in relation to the weld obtained without vibrations was small and amounted to about 25HV0.1. When transverse acoustic vibrations were applied, the hardness also decreased by about 50HV0.1, but at 200 μm below the face, it increased to 350HV0.1. This may indicate a shift deeper the in equipotential surface associated with the heat-affected zone.

SEM investigations have shown significant differences between the structure of areas adjacent to the weld face obtained with the support of acoustic vibrations with a resonance frequency of 1385 Hz and that obtained with a non-vibration weld. The use of lower frequency vibrations led to these significant differences no longer being observed. Acoustic vibrations with a resonant frequency seem to bring about the largest changes in the structure obtained in the face area. Other deeper zones, such as the center, fusing lines or heat-affected zone do not show any greater differences.

The fragmentation of the structure can be achieved by the longitudinal introduction of vibrations inside the pipe, which explains the uniform transverse stresses in the basic pipe material.

The most pronounced directionality of the weld face crystal structure occurs when using vibrations with a resonant frequency, where the direction of crystal growth is perpendicular to the axis of the pipe.

The relationship of the vibration amplitude on the position of the sample in relation to the excitation source was not the subject of research here, although it may have an impact on the obtained effects. However, taking into account the relatively low frequency of acoustic vibrations, their length is many times greater than the distance between the source of vibrations and the welding site used in the experiment. Only in the case of a resonance frequency of 1385 Hz, the wavelength estimated from the simple c/f relationship in the air is approximately 0.24 m. Assuming that the wave leaves the source (transducer, loudspeaker) with the maximum amplitude, it is at a distance of approximately 0.24 m from the source it has only a slightly lower amplitude due to the damping effect (0.07–4 dB/km). Taking into account the much lower frequency of 50 and 110 Hz, we obtain a wavelength in the air of 6.7 and 3 m. It will be different, of course, when the vibrations penetrate into the metal (steel) because the velocity of the longitudinal wave in this case is approximately 5200 m/s and the distances here are key problem.

According to the authors of item [16], who used vibrations with a frequency of 150 to 300 Hz, the main effect on the changes in mechanical properties is their frequency.

On the other hand, the problem of oscillation of the molten metal pool formed by laser welding is undoubtedly very important. Due to the fact that the melt pool vibration tests were not carried out in this case, it is definitely worth referring to what the available publications write on this subject. In one of them [30], model experiments and their comparison with experience were carried out. It has been shown that even very small fluctuations in laser power can lead to strong oscillations in the molten metal pool which is dominated by the natural frequency response. Since the occurring dynamic phenomena are non-linear, bifurcation effects can occur, which tend to expand the natural frequency peaks in the vibration spectrum of the pool. The determined Fourier spectra are in good agreement with the corresponding measurements in the ultraviolet and near infrared ranges during industrial CO_2_ laser welding. The presented model is limited to low beam velocity rates, native materials of low thickness and a small diameter laser beam. This fits well with the experimental conditions used in our article. It can therefore be assumed that the radius of the molten metal pool is small compared to the thickness of the workpiece, as is the case here. Due to the small damping factor of the pool’s radial oscillations, slight laser power fluctuations like 1% at the resonance frequency are sufficient to produce amplitudes large enough for the keyhole to collapse, which can result in a welding imperfections. The main resonant regions of the pool in a 1 mm thick iron element are between 500 and 3500 Hz, depending on the radius of the laser beam and the absorbed laser power. Due to non-linearity, resonance effects can be doubled, or more moderate, at half the natural frequency if the amplitudes are large enough. In addition, each resonance area is extended even by several 100 Hz due to the bifurcation phenomenon. In the case of the TM10 beam mode used, there is an oscillation of the pool, which resembles an ellipse that extends along the beam’s feed direction and shrinks in the perpendicular direction. Although in the quoted item there is no analysis for TEM 10, it can be noticed that assuming the beam speed of 1 m/min, the oscillation frequency of the pool for TEM00 is approximately 230 Hz and for TEM20 approximately 60 Hz.

In another publication [31], it is mentioned that the pool vibration frequency is greater than 1 kHz observed both during spot welding and during welding with a moving beam. In the case of spot welding with a 20 ms laser pulse, oscillations of the weld pool with a frequency in the range of 200–500 Hz were observed, which remained in the first 5 ms after the laser pulse. In the time interval starting at 25 ms and ending about 40 ms from the start of the laser pulse, the oscillation frequency increased to 1.3 kHz. The clotting time was found to be approximately equal to the pulse duration for the spot welding.

In the third position [32], the pool vibration frequency is also close to the value of 200 Hz. It is also indicated that the evolution of the face hump is not a mass movement but a propagation of the phase front which is determined by the alloy ejection towards the sides of the molten metal pool.

In our research, the values of acoustic frequencies of 50, 110 and 1385 Hz were used. Two of them are close to the values quoted in the first and third publications, while one (1385 Hz), being the resonant frequency of the entire laser remelted element, is closer to the value suggested in the second publication. Certainly, it is worth trying to use more frequency values in the 50–2000 Hz range in future research, it would be possible to obtain more complete research. It would be even better to determine numerically, analytically or experimentally the natural frequency of the pool oscillation for a specific TEM mode, power, velocity rate and beam diameter, taking into account other important parameters such as gas flow, material factors, etc. However, this is an issue for a separate publication and it is possible that such attempts will be made.

To sum up, the frequency values used in our research described in this article are in the area close to what were suggested.

In the results obtained here, such dependencies can be noticed, although it is not easy to establish an indisputable trend at this stage of the research. In the future, it is planned to perform research with the use of a greater number of frequency values, so as to capture important relationships.

## 5. Conclusions

A new non-contact method was presented, introducing vibrations into the welded area. Its significant advantages include ensuring constant and repeatable vibration conditions at the welding site and there being no need to ensure a solid assembly of the vibrating system, as opposed to other contact methods of vibration introduction.

SEM and EDX and metallographic testing showed the impact of vibration frequency and direction of their introduction on structures obtained in the surface layer during laser processing. The largest structure changes occurred in the face area of the welds.

The weld dimensions obtained in these experiments were relatively small. In the future, greater beam power should be considered to make the observed changes clearer.

Based on the research, it is not yet possible to clearly determine which direction of vibration or their frequency is the most favorable, because it depends on the expectations. However, we here have a chance for easier application in industrial conditions.

Additional tests in the future in which the frequency and acoustic vibration input angle change more often would allow us to determine the "welding window". However, based on the obtained results, it can be concluded that the proper selection of the direction of vibration introduction and their frequency enables control of the surface layer structure and distribution of alloying elements, which directly affects the properties of the obtained surface layer produced by laser processing with the participation of acoustic vibrations.

## Figures and Tables

**Figure 1 materials-13-04179-f001:**
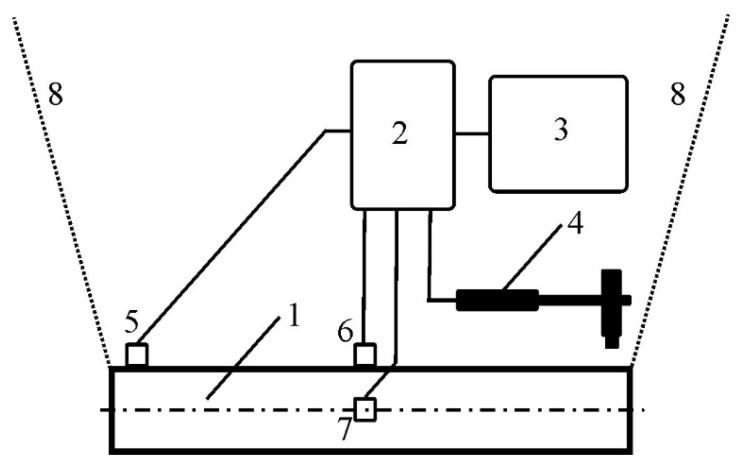
The measuring circuit used to determine the resonant frequency of the pipe: 1—pipe (PM), 2—Bruel and Kjaer 3050-B-060 module, 3—Bruel and Kjaer PULSE Labshop, 4—hammer drill, 5–7—vibration sensors, 8—two suspension ropes.

**Figure 2 materials-13-04179-f002:**
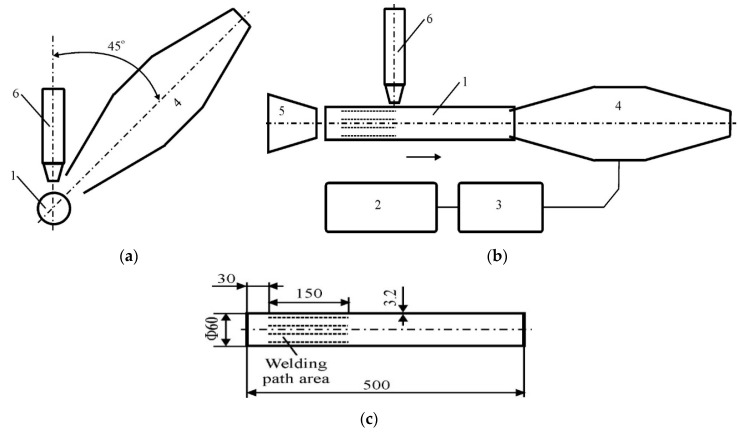
Introduction of acoustic vibrations transversely to the axis of the pipe, perpendicular to its surface (**a**), along the axis of the pipe, into its interior (**b**), area of welding (**c**): 1—pipe dimensions, 2—Bruel and Kjaer type 1023 harmonics generator, 3—the 2734 amplifier, 4—Bruel and Kjaer type 4295 sound source, 5—Bruel and Kjaer type 2236 m, 6—laser head.

**Figure 3 materials-13-04179-f003:**
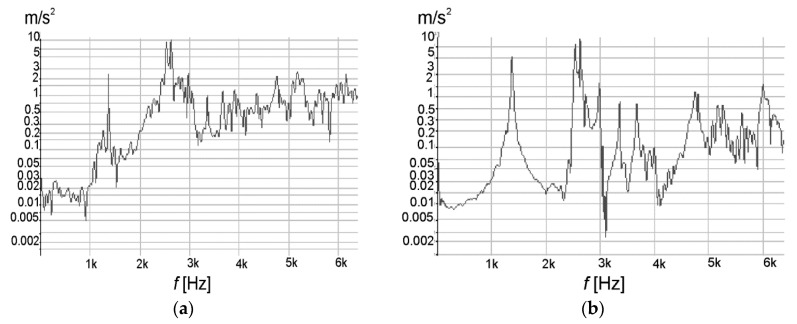
The spectrum of vibration acceleration: (**a**) x-axis and (**b**) y-axis depending on the frequency.

**Figure 4 materials-13-04179-f004:**
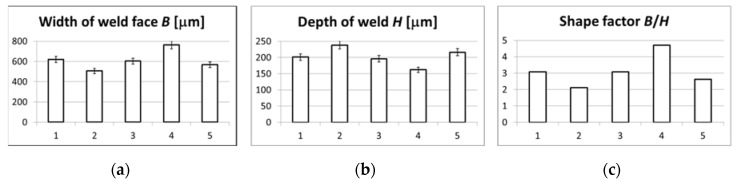
Geometrical factors of five obtained welds: (**a**) width of face *B*, (**b**) depth of weld *H*, (**c**) shape factor *B*/*H.*

**Figure 5 materials-13-04179-f005:**
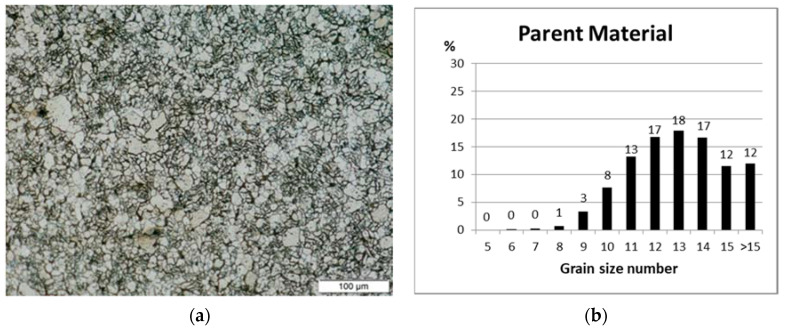
Representative structure and percentage grain distribution of the rectangle area 05/90 μm of: (**a**,**b**) parent material—average grain size 11.58, (**c**,**d**) face of weld No. 4 without vibration—average grain size number 11.02, (**e**,**f**) face of weld No. 1 (50 Hz transverse)—average grain size 11.16, (**g**,**h**) face of weld No. 2 (50 Hz longitudinal)—average grain size 11.26, (**i**,**j**) face of weld No. 3(1385 Hz longitudinal)—average grain size 11.12, (**k**,**l**) face of weld No. 5 (110 Hz longitudinal)—average grain size 10.80, (**m**) summary chart.

**Figure 6 materials-13-04179-f006:**
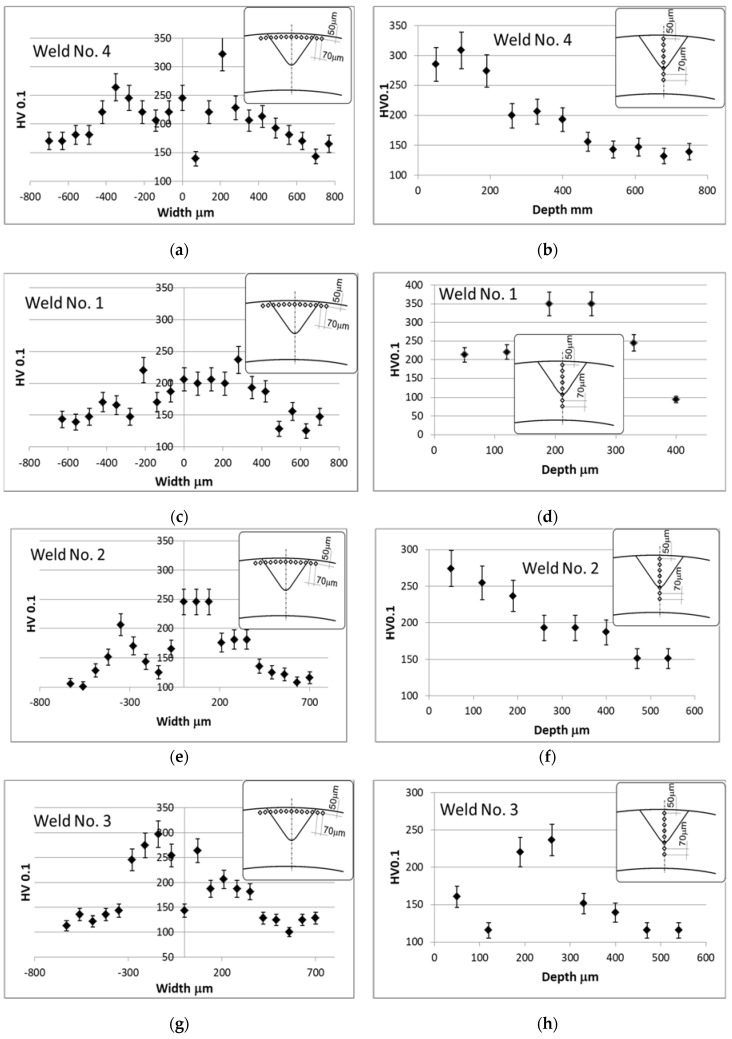
Hardness distribution: Close to the weld face line (**a**,**c**,**e**,**g**,**i**), and in the center of weld at depth direction (**b**,**d**,**f**,**h**,**j**).

**Figure 7 materials-13-04179-f007:**
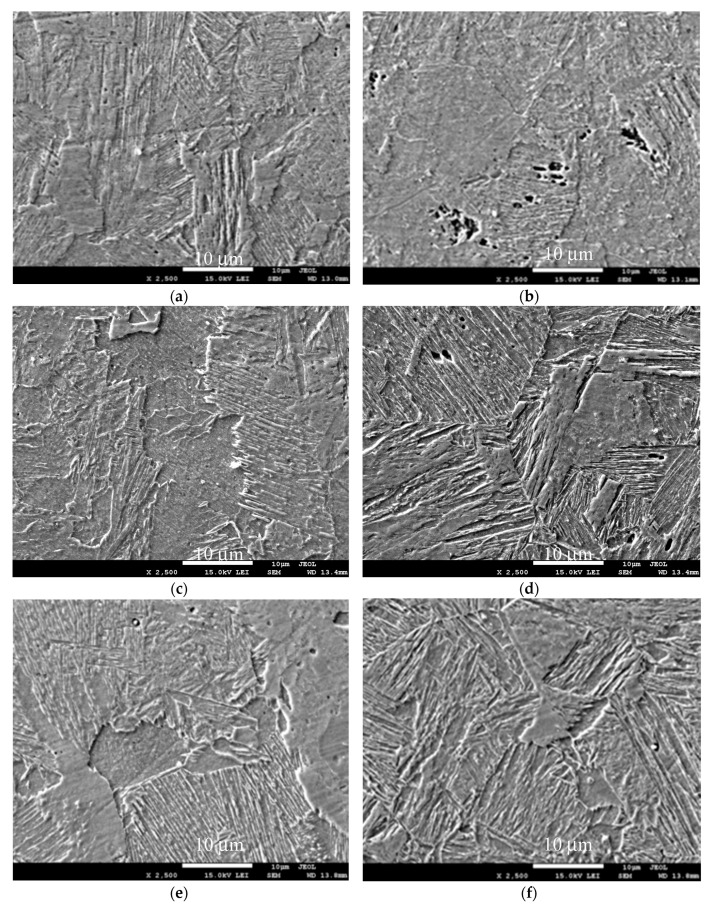
SEM images (2500 ×): in the face zone—(**a**) weld No. 4; (**c**) weld No. 1; (**e**) weld No. 2; (**g**) weld No. 3; (**i**) weld No. 5 and in the center of welds—(**b**) weld No. 4; (**d**) weld No. 1; (**f**) weld No. 2; (**h**) weld No. 3; (**j**) weld No. 5.

**Figure 8 materials-13-04179-f008:**
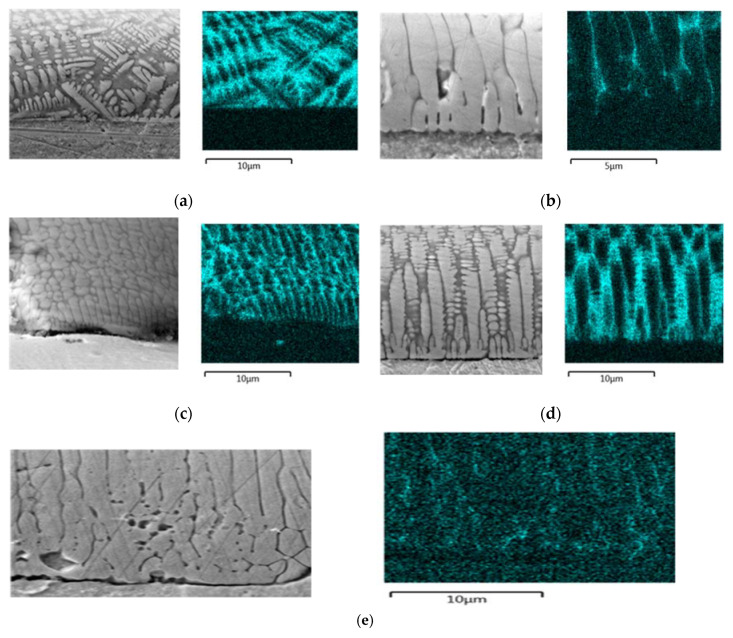
SEM and surface distribution of Si in the face of weld; (**a**) weld No. 4 made without acoustic vibration, (**b**) weld No. 1 (50 Hz with transverse propagation of acoustic vibration), (**c**) weld No. 2 (50 Hz with longitudinal propagation of acoustic vibration), (**d**) weld No. 3 (1385 Hz with longitudinal propagation of acoustic vibration), (**e**) weld No. 5 (110 Hz with longitudinal propagation of acoustic vibration).

**Table 1 materials-13-04179-t001:** Chemical composition and properties of P235GH steel according to PN-EN 10204.

C [%] 0.11	Mn [%] 0.42	Si [%] 0.016	P [%] 0.008	S [%] 0.006	Cu [%] 0.02	Cr [%] 0.02	Ni [%] 0.01	Al [%] 0.03	Al_sol_ [%] 0.025
V [%] 0.001	Mo [%] 0.002	Nb [%] -	Co [%] 0.002	Ti [%] 0.001	As [%] 0.001	N_2_ [%] 0.007	Ca [%] 0.0026	Pb [%] 0.001	Sn [%] 0.001
O [%] -	H_2_ [%] -	Zn [%] 0.0012	W [%] 0.001	B [%] 0.0002	Zr [%] 0.0006	*C_EV_* [%] 0.19	*R_e_* [MPa] 311	*R_m_* [MPa] 408	*A_5_* [%] 45.6

**Table 2 materials-13-04179-t002:** Welding parameters and specification regarding the introduced vibrations: ***f***—acoustic frequency [Hz], ***P***—power of the laser [W], ***v***—welding speed [m/min], direction of vibrations introduction according to Figure 2a–c.

Weld No.	*F* [Hz]	*P* [W]	*V* [m/min]	Direction of Vibration
1	50	1150	0.5	transverse
2	50	1150	0.5	longitudinal
3	1385	1150	0.5	longitudinal
4	-	1150	0.5	-
5	110	1150	0.5	longitudinal

**Table 3 materials-13-04179-t003:** Parameters during tests. ***f***—vibrations frequency, ***P***—power of the laser, Direction—direction of vibrations introduction, ***B***—face width, ***H***—depth of penetration, ***B***/***H***—weld shape factor.

Weld No.	*F* [Hz]	*P* [W]	Direction	*B* [µm]	*H* [µm]	*B*/*H*
1	50	1150	transverse	619	201	3.08
2	50	1150	longitudinal	505	238	2.12
3	1385	1150	longitudinal	603	196	3.08
4	-	1150	-	762	162	4.70
5	110	1150	longitudinal	568	216	2.63

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
