# Peer review of "A Novel Method of Supporting the Laser Welding Process with Mechanical Acoustic Vibrations"

_materials, 2020, doi:10.3390/ma13184179_

Round 1
Reviewer 1 Report
- Massive clustering of references, e. g. page 1,- line 27, 1-36 must be avoided
- page 1, line 27/28: "The issue of improving the structure or
28 properties can generally be interpreted as improving weldability, which has always been a priority." --> please explain
- page 1, line 29 following: references are missing
- page 1, line 38 following: references are missing
- Introduction: a clear assignment of the literature references to the named aspects is impossible due to the clustering of references. The structure needs to be revised. Several times it is stated that a certain method is used ("One such method is presented in this article."), then further explanations follow, then it is stated again that a certain method is used. Which method? When it’s sound I would expect to find some more information in the state of the art.
- page 3, line 104 following: “it was decided to use a laser source, which 105 by its nature guarantees stable and repeatable welding of the substrates.” – That's popular science.
- page 3, line 110: How was the shielding gas supplied?
- page 3, line 112 following: Why have you chosen the named frequencies? When its about the resonant frequency. Wouldn’t it be better to address the resonant frequency of the molten pool instead of the complete structure?
- page 4, Table 2: Where do the parameters come from and why were they chosen? The welding speeds are very, very slow for a laser beam process. This should also result in a very wide weld pool - how transferable are the results to other (more realistic) welding speeds?
- Chapter 3.1 / Figure 3: The axis labeling is unreadable. The explanations cannot be accepted on this basis.
- Chapter 3.2: Size of the sample for statistics? In contrast to the claim that laser beam welding is so stable, there are usually fluctuations in the welding depth - this must be shown. Microsections of the whole weld should be shown, whether more effects occur.
- Chapter 3.3: The comparability of the grain size measurement is questionable - was a comparable image section always chosen? First the weld metal is recorded, then also the heat-affected zone. What is the sample size regarding statistics? The discussion of the effects is missing!
- Figure 5, a-l: Please add the parameters instead of Weld No. I
- Figure 6: Labeling in sketches unreadable.
- Page 9, line 195 following: a discussion is missing!
- Figure 7: Labels and scales are unreadable. The numbering is confusing, parameters are missing.
- Chapter 4: The discussion is clearly brief for clarity, effects are named, but their causes are not sufficiently described or discussed. The systematic elaboration of effects, e.g. by a specific investigation of further frequencies, is missing.
Author Response
Dear Reviewer,
below I have addressed your comments in the order in which they appeared. I applied corrections to the version linguistically corrected by the Publisher. Therefore, there may be some minor changes to the text. I have attached the corrected version of the article to the MDPI Materials system.
Comments and Suggestions for Authors
- Massive clustering of references, e. g. page 1,- line 27, 1-36 must be avoided
Re:
Thank you very much for your time and critical review. I tried to express my answer on each point in turn what is happening below. I also attach a revised version of the article, in which I took into account Your comments and from the second Reviewer. I also made small corrections that I received from the English correction desk of the text from the Publisher.
Mass grouping of literature items has been restructured and the current citations relate to the issues raised in the text. Some citations will be cited later concerning the local context.
- page 1, line 27/28: "The issue of improving the structure or 28 properties can generally be interpreted as improving weldability, which has always been a priority." --> please explain
Re:
Indeed, it was not clear what weldability the comment was about and the final statement was too categorical. Therefore, I propose the following sentence: “The issue of improving the micro and macro structural properties can generally be interpreted as improving metallurgical weldability, which has often been a priority”
- page 1, line 29 following: references are missing
Re:
The citation of the relevant item to the issue was included.
- page 1, line 38 following: references are missing
Re:
The citation of the relevant item to the issue was included.
- Introduction: a clear assignment of the literature references to the named aspects is impossible due to the clustering of references. The structure needs to be revised. Several times it is stated that a certain method is used ("One such method is presented in this article."), then further explanations follow, then it is stated again that a certain method is used. Which method? When it’s sound I would expect to find some more information in the state of the art.
Re:
The structure of the introduction has been revised so as to clearly cite the items of literature where necessary. I also cited relevant literature items relating to the applied technical solutions.
- page 3, line 104 following: “it was decided to use a laser source, which 105 by its nature guarantees stable and repeatable welding of the substrates.” – That's popular science.
Re:
The intention of this formulation was to draw attention to the use of the advantages of laser welding in terms of the accuracy and repeatability of head or table movements, as well as the insensitivity of the laser beam to the presence of vibrations, as opposed to arc welding methods.
- page 3, line 110: How was the shielding gas supplied?
Re:
In the experiments, an argon blast in the beam axis was used with a capacity of 10 l/min through a nozzle 4 mm in diameter spaced from the parent material surface by 5 mm. I have put this information in the appropriate place in the chapter No. 2 (line: 119)
- page 3, line 112 following: Why have you chosen the named frequencies? When its about the resonant frequency. Wouldn’t it be better to address the resonant frequency of the molten pool instead of the complete structure?
Re:
Due to the fact that the area of impact of the applied acoustic wave is much larger than the weld pool, it was assumed that taking into account the resonant vibrations of the entire tested object would be more adequate here. Several problems are encountered in trying to determine the resonant frequency of the weld pool. These include high metal temperature, small volume, and a short lifetime of the liquid phase. It is worth planning the experiment in the future so that, as part of determining the boundary conditions, model or analytically estimate the resonance frequency of the molten metal pool and then apply it by introducing vibrations into the welding zone.
At the initial stage, which is the research described in the article, only the values from the upper and lower frequency range were used to emphasize the possible differentiation of effects.
It was decided to choose such frequencies that they would be possible to obtain with the use of the existing acoustic system (limitations resulting from the frequency response of the loudspeaker used).
- page 4, Table 2: Where do the parameters come from and why were they chosen? The welding speeds are very, very slow for a laser beam process. This should also result in a very wide weld pool - how transferable are the results to other (more realistic) welding speeds?
Re:
Due to the fact that the tests were initial stage of research, the parameters were selected with the desire to observe the effect of vibrations of different frequencies on the solidifying pool, which is easier to see when we are dealing with a larger volume of the molten metal pool.
Due to the high dynamics of the process, transferring the results to higher speeds will be possible after additional tests with specific welding speeds are performed and may be the next stage of work in this area.
- Chapter 3.1 / Figure 3: The axis labeling is unreadable. The explanations cannot be accepted on this basis.
Re:
The font was enlarged when describing the axes in the charts.
- Chapter 3.2: Size of the sample for statistics? In contrast to the claim that laser beam welding is so stable, there are usually fluctuations in the welding depth - this must be shown. Microsections of the whole weld should be shown, whether more effects occur.
Re:
The results published in the article regarding the shape factors are the arithmetic mean of the 3 cross sections. I will provide information about it in the article in the appropriate place. (line: 176)
The results of the shape coefficient tests were performed each time for cross-sections distant from each other by 5 mm. In the course of metallographic tests, the differences in the measured values were within the range of +/- 10mm (line: 180)
- Chapter 3.3: The comparability of the grain size measurement is questionable - was a comparable image section always chosen? First the weld metal is recorded, then also the heat-affected zone. What is the sample size regarding statistics? The discussion of the effects is missing!
Re:
The area where the grain size was measured was limited to rectangle with sides 555/416 µm. The results presented in the graphs are the average of three measurements for each weld. In the text described the results (line: 209) and add summary chart (Fig. 5m) concerning to this topic.
- Figure 5, a-l: Please add the parameters instead of Weld No. I
Re:
The parameters for which subsequent welds were added to the description of Figure 5.
- Figure 6: Labeling in sketches unreadable.
Re:
Improved visibility of the labeling of graphs in Figure 6.
- Page 9, line 195 following: a discussion is missing!
Re:
A discussion of the obtained results of hardness measurement has been added (line: 242).
- Figure 7: Labels and scales are unreadable. The numbering is confusing, parameters are missing.
Re:
The Figure 7 with SEM test were enlarged, thus improving the visibility of the test parameters.
- Chapter 4: The discussion is clearly brief for clarity, effects are named, but their causes are not sufficiently described or discussed. The systematic elaboration of effects, e.g. by a specific investigation of further frequencies, is missing.
Re:
After the research, one thing can be said for sure. The main cause of the changes occurring is the cyclical change of the acoustic pressure, which generate transverse stresses in the pipe (when it comes to introducing acoustic vibrations into pipe) and the direct cyclical change of the acoustic pressure of vibrations supplied from the outside transversely to the weld axis. I agree that confirmation of this requires further systematic trials with more frequency values and greater vibration amplitude values.
Thank you very much for your thorough criticism of our article. I hope that, at least partially, we managed to correct the shortcomings, which will make the article better reflect the obtained results and will allow for better systemic planning of further research in the future.

Reviewer 2 Report
The paper demonstrates that different microstructural properties might be achieved during the laser welding if vibrations of workpiece are induced contactless using high-pressure sound source. The method is interesting since it is much more practical compared to contact inducement of vibrations and could be in principle well integrated into a laser welding head. However, I suggest some major corrections in order to improve the paper before being accepted:
- The title and conclusions suggest, that the method is effective and that significant advantages are presented in the paper, such as “constant and repeatable conditions of vibration at the welding site…”, but I have not found any experimental results, which would prove that statement. How the amplitude and vibrations spectra is changing during welding? How the vibration amplitude depends on the position of the specimen relative to the excitation source? How repeatable is the welding process using acoustic vibration?
- The introduction section is written in a very general form. I miss more detailed summary of the individual reference. Which papers describe the most relevant research to the contactless inducing of vibrations during laser welding – to this work?
- An interesting way of laser-induced vibrations is mentioned, however ref. 38 is not relevant. Check for lapsus. I recommend reading of recently published paper (Matej Senegačnik, Matija Jezeršek, Peter Gregorčič, "Propulsion effects after laser ablation in water, confined by different geometries", Applied physics. A, Materials science & processing, Feb. 2020, vol. 126, iss. 2, str. 1-12, ilustr., ISSN 0947-8396, https://link.springer.com/article/10.1007%2Fs00339-020-3309-y, DOI: 1007/s00339-020-3309-y.) which explains the principles of laser excitation of vibrations.
- Induced vibrations should be measured as a movement of surface in the vicinity of the melt pool in all three directions (out-of-plane, in-plane longitudinal and in-plane perpendicular). The measure should be an amplitude of harmonic movement at the excitation frequency. Only in this way other workpiece geometries and materials could be compared to each other in further research.
- There is no discussion on physical mechanisms that drive the differences between welding under vibrations and without them. Is perhaps only the difference in laser beam path geometry, which is like wobbling (see for example https://www.wlt.de/lim/Proceedings2017/Data/PDF/Contribution236_final.pdf) due to oscillatory movement of the specimen?
Minor comments:
- Laser beam diameter at the workpiece surface must be included.
- From results is evident that the laser conduction welding regime was used, which is less favorable compared to the deep-penetration. Could you explain why you decided to use this regime?
Author Response
Dear Reviewer,
below I have addressed your comments in the order in which they appeared. I applied corrections to the version linguistically corrected by the Publisher. Therefore, there may be some minor changes to the text. I have attached the corrected version of the article to the MDPI Materials system.
Comments and Suggestions for Authors
The paper demonstrates that different microstructural properties might be achieved during the laser welding if vibrations of workpiece are induced contactless using high-pressure sound source. The method is interesting since it is much more practical compared to contact inducement of vibrations and could be in principle well integrated into a laser welding head. However, I suggest some major corrections in order to improve the paper before being accepted:
The title and conclusions suggest, that the method is effective and that significant advantages are presented in the paper, such as “constant and repeatable conditions of vibration at the welding site…”, but I have not found any experimental results, which would prove that statement. How the amplitude and vibrations spectra is changing during welding? How the vibration amplitude depends on the position of the specimen relative to the excitation source? How repeatable is the welding process using acoustic vibration?
The introduction section is written in a very general form. I miss more detailed summary of the individual reference. Which papers describe the most relevant research to the contactless inducing of vibrations during laser welding – to this work?
An interesting way of laser-induced vibrations is mentioned, however ref. 38 is not relevant. Check for lapsus. I recommend reading of recently published paper (Matej Senegačnik, Matija Jezeršek, Peter Gregorčič, "Propulsion effects after laser ablation in water, confined by different geometries", Applied physics. A, Materials science & processing, Feb. 2020, vol. 126, iss. 2, str. 1-12, ilustr., ISSN 0947-8396, https://link.springer.com/article/10.1007%2Fs00339-020-3309-y, DOI: 1007/s00339-020-3309-y.) which explains the principles of laser excitation of vibrations.
Induced vibrations should be measured as a movement of surface in the vicinity of the melt pool in all three directions (out-of-plane, in-plane longitudinal and in-plane perpendicular). The measure should be an amplitude of harmonic movement at the excitation frequency. Only in this way other workpiece geometries and materials could be compared to each other in further research.
There is no discussion on physical mechanisms that drive the differences between welding under vibrations and without them. Is perhaps only the difference in laser beam path geometry, which is like wobbling (see for example https://www.wlt.de/lim/Proceedings2017/Data/PDF/Contribution236_final.pdf) due to oscillatory movement of the specimen?
Re:
Thank you very much for your thorough evaluation of our article. Agree that the word "effective" can be understood as being absolutely effective. This was not the original intention. The point was that the proposed method of introducing acoustic vibrations turned out to be effective in terms of determining structural changes and mechanical properties of the obtained joints. If the title of the article is perceived this way, it may remove the adjective "effective".
The fragment of the sentence "constant and repeatable vibration conditions at the welding site ..." refers here only to the aspect of instability, such as, for example, in welding with an electric arc.
The spectrum of amplitude and vibration during welding was not measured, only the acoustic pressure was measured, which was in each case L = 100dB. This value can of course be converted into a stress p in Pa. In this case, we get p = 2 * 10-5 * 105L/20 = 2 Pa (I will put this value in the text of the article). It is not a large value, however, it must be remembered that the pressure changes cyclically and this affects the welding results. The dependence of the vibration amplitude on the position of the sample in relation to the excitation source was not the subject of research here, although it may have an impact on the obtained effects. However, taking into account the relatively low frequency of acoustic vibrations, their length is many times greater than the distance between the source of vibrations and the welding site used in the experiment. Only in the case of a resonance frequency of 1385 Hz, the wavelength estimated from the simple c/f relationship in the air is approx. 0.24 m. Assuming that the wave leaves the source (transducer, loudspeaker) with the maximum amplitude, it is at a distance of approx. 0.24 m from the source it has only a slightly lower amplitude due to the damping effect (0.07-4 dB/km). Taking into account the much lower frequency of 50 and 110 Hz, we obtain a wavelength in the air of 6.7 and 3 m. It will be different, of course, when the vibrations penetrate into the metal (steel) because the velocity of the longitudinal wave in this case is approx. 5200 m / s and the distances here are key problem. Thank you for paying attention to this issue. I think that it is worth developing these considerations in the future and planning experiments in such a way as to investigate the effect of the distance of the welding zone from the source of acoustic vibrations.
The repeatability of the laser welding process with the use of acoustic vibration is very important due to the credibility of the experiment results, but also possible application in the industry will be considered in further research. Thank you for pointing this out.
Possible deliberations on the mechanisms that are responsible for the effects obtained during laser welding with the participation of acoustic vibrations would be, I think, premature or incomplete at this stage, although it can already be concluded that the key role will be played by cyclical stresses in the parent material (in the case when introducing vibrations axially inside the pipe) and directly cyclic sound pressure of vibrations (when the vibrations reach the weld zone from the outside approximately perpendicular to the axis of the pipe). In future research, I intend to carry out some additional experiments with different frequency values, amplitudes and different directions of their introduction during laser welding to get an answer which parameters of the technological regime or additional effects that you have pointed out to me, such as beam wobble, may be crucial for the results obtained. I have read the featured articles. Both are very interesting and I think they will help me plan the methodology for further trials and tests.
In line 66 (after revise 71), I referred to a significantly erroneous position regarding laser vibration generation. In the corrected text that I am providing it has been corrected - it should be in the [3,28]
Minor comments:
Laser beam diameter at the workpiece surface must be included.
From results is evident that the laser conduction welding regime was used, which is less favorable compared to the deep-penetration. Could you explain why you decided to use this regime?
Re:
Thank you for drawing my attention to the parameters describing the laser beam and its focusing relative to the surface. Indeed, they are very important due to the interpretation of the obtained results. A beam of coherent light was used, which was 1 mm in diameter on the surface of the workpiece and in the TEM 10 mode. As the focus (d = 0.4 mm) of the beam fell above the surface, it was impossible to obtain a steam channel. I added this information in Chapter 2 (line 118 follow).This was dictated by the desire to obtain a larger size of the metal pool, thus allowing for easier observation of changes in its structure after crystallization.
Thank you for your constructive criticism of our article and I think it will make it better quality. A deeper assessment of the mechanisms cannot be made at this stage. In the future, attempts to investigate changes in, for example, the crystallographic orientation may be considered.

Round 2
Reviewer 1 Report
Despite additions and improvements, the contribution has significant weaknesses:
- no statistical validation. In my opinion, the results are not so clear that they can be verified without a higher sample size and are sufficient for publication in "materials".
- it is postulated that the frequency of the overall structure is decisive, whereas other authors in the state of the art for frequency superposition directly address the molten pool - which seems to make more sense due to the aim of the solidification processes taking place there. At least the comparison with the selected frequencies and the natural frequency of the melt pool should be carried out to support the own hypothesis.
- no comparability in the data collected for optical microscopy. If the rectangle is 555xx416 micron², that is good to know - but the measuring position is still not comparable between the samples. I have measured on the basis of the microsections and in doing so the heat-affected zone is partly taken into account and partly not. This is not comparable.
- The massive clustering of literature still exists and thus does not provide a comprehensible state of the art
Author Response
- no statistical validation. In my opinion, the results are not so clear that they can be verified without a higher sample size and are sufficient for publication in "materials".
Re:
Graphs showing the results of measurements of geometric parameters (Fig. 4) and the results of hardness tests (Fig. 6) were supplemented by adding the standard deviation for each measuring point. Indeed, the sample size is not large, but planned future studies will take this into account.
- it is postulated that the frequency of the overall structure is decisive, whereas other authors in the state of the art for frequency superposition directly address the molten pool - which seems to make more sense due to the aim of the solidification processes taking place there. At least the comparison with the selected frequencies and the natural frequency of the melt pool should be carried out to support the own hypothesis.
Re:
The problem of oscillation of the molten metal pool formed by laser welding is undoubtedly very important. Due to the fact that the melt pool vibration tests were not carried out in this case, it is definitely worth referring to what the available publications write on this subject. In one of them (T Klein, M Vicanek and G Simon, Forced oscillations of the keyhole in penetration laser beam welding, J. Phys. D: Appl. Phys. 29 (1996) 322–332.), model experiments and their comparison with experience were carried out. It has been shown that even very small fluctuations in laser power can lead to strong oscillations in the molten metal pool which is dominated by the natural frequency response. Since the occurring dynamic phenomena are non-linear, bifurcation effects can occur, which tend to expand the natural frequency peaks in the vibration spectrum of the pool. The determined Fourier spectra are in good agreement with the corresponding measurements in the ultraviolet and near infrared ranges during industrial CW CO2 laser welding. The presented model is limited to low beam velocity rates, native materials of low thickness and a small diameter laser beam. This fits well with the experimental conditions used in our article. It can therefore be assumed that the radius of the molten metal pool is small compared to the thickness of the workpiece, as is the case here. Due to the small damping factor of the pool's radial oscillations, slight laser power fluctuations like 1% at the resonance frequency are sufficient to produce amplitudes large enough for the keyhole to collapse, which can result in a welding imperfections. The main resonant regions of the pool in a 1 mm thick iron element are between 500 and 3500 Hz, depending on the radius of the laser beam and the absorbed laser power. Due to non-linearity, resonance effects can be doubled, or more moderate, at half the natural frequency if the amplitudes are large enough. In addition, each resonance area is extended even by several 100 Hz due to the bifurcation phenomenon. In the case of the TM10 beam mode used, there is an oscillation of the pool, which resembles an ellipse that extends along the beam's feed direction and shrinks in the perpendicular direction. Although in the quoted item there is no analysis for TEM 10, it can be noticed that assuming the beam speed of 1 m/min, the oscillation frequency of the pool for TEM00 is approx. 230 Hz and for TEM20 approx. 60 Hz.
In another publication (VV Semak, JA Hopkins, MH McCay and TD McCay, Melt pool dynamics during laser welding Journal of Physics D: Applied Physics, Volume 28, Number 12, 1995) it is mentioned that the pool vibration frequency is greater than 1 kHz observed both during spot welding and during welding with a moving beam. In the case of spot welding with a 20 ms laser pulse, oscillations of the weld pool with a frequency in the range of 200-500 Hz were observed, which remained in the first 5 ms after the laser pulse. In the time interval starting at 25 ms and ending about 40 ms from the start of the laser pulse, the oscillation frequency increased to 1.3 kHz. The clotting time was found to be approximately equal to the pulse duration for the spot welding.
Also in the third position (Mickael Courtois, Muriel Carin, Philippe Le Masson, Sadok Gaied, Mikhaël Balabane. A complete model of keyhole and melt pool dynamics to analyze instabilities and collapse during laser welding. Journal of Laser Applications, Laser Institute of America, 2014 , 26 (4), pp. 042001. 10.2351 / 1.4886835), the pool vibration frequency is close to the value of 200 Hz. It is also indicated that the evolution of the face hump is not a mass movement but a propagation of the phase front which is determined by the alloy ejection towards the sides of the molten metal pool.
In our research, the values of acoustic frequencies of 50, 110 and 1385 Hz were used. Two of them are close to the values quoted in the first and third publications, while one (1385 Hz), being the resonant frequency of the entire laser remelted element, is closer to the value suggested in the second publication. Certainly, it is worth trying to use more frequency values in the 50-2000 Hz range in future research, it would be possible to obtain more complete research. It would be even better to determine numerically, analytically or experimentally the natural frequency of the pool oscillation for a specific TEM mode, power, velocity rate and beam diameter, taking into account other important parameters such as gas flow, material factors, etc. However, this is an issue for a separate publication and it is possible that such attempts will be made.
To sum up, the frequency values used in our research described in this article are in the area close to what were suggested.
- no comparability in the data collected for optical microscopy. If the rectangle is 555xx416 micron², that is good to know - but the measuring position is still not comparable between the samples. I have measured on the basis of the microsections and in doing so the heat-affected zone is partly taken into account and partly not. This is not comparable.
Re:
The surface area used for the study of the grain size is indeed large and in some cases it happens that both the weld, the heat-affected zone or the native material are included in it. We decided to perform re-measurements of the grain size in the selected areas near the face of the obtained joints. In the photographs presented in Figure 5, areas in the form of a rectangle with dimensions of 105 / 90 mikron each are marked. Now, the research results posted should be more representative of the specific areas mentioned above.
- The massive clustering of literature still exists and thus does not provide a comprehensible state of the art.
Part of the literature has been removed. Only those necessary to discuss the state of the art remained.
Reviewer 2 Report
Authors have sufficiently improved the manuscript. I have no further comments.
Author Response
Thank you very much for your valuable comments.